# Effects of Different Photoperiods on the Transcriptome of the Ovary and Small White Follicles in Zhedong White Geese

**DOI:** 10.3390/ani14182747

**Published:** 2024-09-23

**Authors:** Tao Huang, Meina Fei, Xiaolong Zhou, Ke He, Songbai Yang, Ayong Zhao

**Affiliations:** Key Laboratory of Applied Technology on Green-Eco-Healthy Animal Husbandry of Zhejiang Province, College of Animal Science and Technology, College of Veterinary Medicine, Zhejiang A&F University, Hangzhou 311300, China; taohuang@zafu.edu.cn (T.H.); 16604565698@163.com (M.F.); zhouxiaolong@zafu.edu.cn (X.Z.); heke@zafu.edu.cn (K.H.); sbyang@zafu.edu.cn (S.Y.)

**Keywords:** Zhedong white goose, photoperiod, RNA-seq analysis, ovary, small white follicles

## Abstract

**Simple Summary:**

Birds and poultry are seasonal breeding animals, and photoperiod is an important environmental factor affecting their broodiness. The Zhedong white goose has a strong broodiness, and extending the photoperiod can improve its egg-laying performance. Photoperiod can advance the sexual maturity of poultry and accelerate the development of reproductive organs and follicles. The growth and development of the ovary and follicles play a major role in the ability of a goose to produce eggs. Therefore, this paper investigates the effects of different photoperiods on the ovaries and small white follicles in the Zhedong white goose. Transcriptome sequencing identified differentially expressed genes and key gene modules, and revealed the effect of photoperiod on gene expression of ovaries and small white follicles. Functional enrichment analysis showed that photoperiod could affect smooth muscle cell proliferation in ovaries, as well as extracellular matrix-related function in small white follicles. In conclusion, the results provide a more comprehensive insight into the molecular mechanism through which photoperiod influences reproduction in geese.

**Abstract:**

Photoperiod can regulate the broodiness of geese and thus increase their egg-laying rate. The laying performance of geese is mainly determined by ovary and follicle development. To understand the effect of photoperiod on the ovary and small white follicles, sixteen 220-day-old healthy female Zhedong white geese were randomly divided into two groups for long photoperiods (15L:9D) and short photoperiods (9L:15D). The geese were euthanized after two months of feeding, and their ovaries and follicles were collected for transcriptome sequencing. RNA-seq analysis identified 187 and 448 differentially expressed genes in ovaries and small white follicles of different photoperiod groups, respectively. A long photoperiod promotes high expression of *SPP1*, *C6*, *MZB1*, *GP1BA,* and *FCGBP* genes in the ovaries, and increases the expression of *SPP1*, *ANGPTL5*, *ALPL*, *ZP1,* and *CHRNA4* genes in small white follicles. Functional enrichment analysis showed that photoperiod could affect respiratory system development, smooth muscle cell proliferation in ovaries, and extracellular matrix-related function in small white follicles. WGCNA revealed 31 gene modules, of which 2 were significantly associated with ovarian weight and 17 with the number of small white follicles. Our results provide a better understanding of the molecular regulation in the photoperiod affecting goose reproduction.

## 1. Introduction

Poultry are seasonal breeding animals, and light is an important environmental factor that affects their growth, development, and reproductive performance [1]. Seasonal variation in reproductive activity varies among avian species, and the effects of light on their reproductive endocrinology also vary. The potential benefits of photoperiod on bird reproduction have been widely reported, including earlier sexual maturity and increased reproductive organ and follicle development [2,3,4,5,6]. Geese are a seasonal breeding species, and light affects how well they lay their eggs. Many Chinese goose breeds have strong broodiness, with egg-laying numbers per year ranging from 24 to 70. Extending the photoperiod can improve the egg-laying performance of geese [7,8]. 

The growth and development of the ovary and follicles play a major role in the ability of a goose to produce eggs. Follicle formation begins when the goose achieves sexual maturity. There are a lot of follicles in the follicular pool that are less than one millimeter in diameter. Most of these follicles begin to grow and are sorted into graded follicles through screening. As they continue to develop, graded follicles start the ovulation process. Graded follicles further mature and enter the ovulation process. Pre-graded follicles include small white follicles (SWFs), large white follicles (LWFs), and small yellow follicles (SYFs) [9]. Atresia most frequently occurs in developing follicles that are not selected to finish follicle development [10]. Pre-graded follicles are more susceptible to follicular atresia compared to graded follicles, and the number of atretic follicles is inversely correlated with poultry egg-laying performance. As a result, the follicle selection process is critical for the orderly growth of follicles [11]. Avian light stimulation transmits signals to the hypothalamus, which releases hormones via the hypothalamic–pituitary–gonadal axis (HPGA) that regulate ovarian and follicular development. According to the study’s results, both absolute and relative ovarian weights increased with increasing light intensity [12]. The growth of follicles is improved, and poultry lay better when the number and weight of follicles are larger [13,14,15].

Recently, transcriptome sequencing technology (RNA-seq) and its new biological applications have advanced rapidly [16]. It has become increasingly convenient to probe the molecular mechanisms of animal physiological responses at the genomic level. RNA-seq analysis and bioinformatics were used to compare pituitary mRNA expression patterns in Zhedong white geese (*Anser cygnoides*) during different seasons and identified *GnRHR* and *DIO2* as candidate genes for the molecular regulation of breeding seasonality [17]. RNA-seq analysis validates the role of follicle-stimulating hormone-mediated signaling pathways and key response motifs in goose granulosa cell proliferation [18].

The Zhedong white goose is a domestic Chinese breed with a high tendency of broodiness and a low egg-laying capacity [19]. The Zhedong white goose only lays a total of 30-35 eggs during the entire breeding season [19]. The egg-laying period of geese can be prolonged, or the counter-seasonal reproduction of geese can be regulated by controlling the light time, which in turn improves the egg-laying performance of geese. In this study, we used RNA-seq analysis and Weighted Correlation Network Analysis (WGCNA) to screen key genes and pathways affecting ovarian and follicular development under different photoperiods, and to explore the regulatory role of photoperiods on ovaries and SWF development in geese.

## 2. Materials and Methods

### 2.1. Animal Experimental Design and Sample Collection

The Zhedong white geese used in this study were raised on the Zhedong white goose breeding farm (XiangShan, Zhejiang, China). A total of sixteen 220-day-old healthy female Zhejiang white geese with similar body weight were randomly divided into 2 groups, with eight geese in each group. The experiment was divided into two groups: the long-photoperiod (LP) group with 15 h of continuous light and 9 h of darkness (15L:9D) and the short-photoperiod (SP) group with 9 h of continuous light and 15 h of darkness (9L:15D). Geese were raised in artificial light with a controlled intensity of 30 Lux, in accordance with standard feeding and management procedures. The geese were euthanized after two months of feeding. Thereafter, the ovaries and follicles were collected, and the ovaries were weighed. According to the follicle diameter, the follicles of different grades were categorized and counted. Student’s *t*-test was used to analyze the differences between LP and SP groups in ovarian weight and number of small white follicles, respectively. As soon as the samples were taken, they were frozen in liquid nitrogen and stored at a temperature of −80 °C. The ovaries and small white follicles of a total of six geese were subjected to transcriptome sequencing. The six geese were randomly selected from the LP and SP groups, with three geese randomly selected from each group of eight geese, respectively.

### 2.2. RNA Extraction, Library Construction, and Sequencing

Total RNA was extracted from the ovaries and SWFs using TRIzol reagent (Invitrogen, Hong Kong, China) according to the manufacturer’s instructions. RNA quality (RNA integrity number, RIN) was determined using an Agilent 2100 Bioanalyser and quantified using the ND-2000 (NanoDrop Technologies, Wilmington, DE, USA). High-quality RNA samples (OD260/280 ≥ 1.8, OD260/230 ≥ 1.0, RIN ≥ 6.5, ≥1 μg) were selected for library construction and Illumina RNA-seq. A-T base pairing was performed with poly (A) mRNA using magnetic beads with Oligo (dT), and small fragments of about 300 bp were screened by adding fragmentation buffer. First-strand cDNA was synthesized using mRNA as a template with the addition of reverse transcriptase and random hexamers, followed by second-strand cDNA synthesis. Then, 2% Low-Range Ultra Agarose was screened to obtain about 300 bp cDNA target fragments, and constructed libraries were amplified by PCR for 15 cycles. After quantification by Qubit 4.0, the libraries were sequenced using Illumina HiSeq PE 2X150 bp (Illumina, San Diego, CA, USA) read length. RNA-seq was performed by Shanghai Majorbio Bio-pharm Biotechnology Co., Ltd. (Shanghai, China). 

### 2.3. RNA-seq Data Analysis

The raw reads were filtered using fastp v 0.19.5 [20], and the average error rate of the sequencing bases corresponding to quality control data was below 0.1%. The Q20, Q30, and GC contents of clean reads were calculated. Clean reads were separately aligned to the Zhedong white goose (*Anser cygnoides*) reference genome (GCF_000971095.1) using HISAT2 v.2020 [21]. The mapped reads were assembled and spliced using the software StringTie v1.3.3b to obtain transcripts using a reference-based approach [22]. To identify differentially expressed genes (DEGs) between the two groups, the expression level for each transcript was calculated using the fragments per kilobase of exon per million mapped reads (FPKM) method. The differential gene expression analysis was performed using DESeq2 v 1.24.0 [23]. Genes with false discovery rate (FDR) < 0.05 and |log2 (fold change)| >1 were screened as differentially expressed genes.

### 2.4. Enrichment Analysis and Protein–Protein Interaction Network Analysis

The functional enrichment analysis was performed using the Gene Ontology (GO) and Kyoto Encyclopedia of Genes and Genomes (KEGG) databases to identify which DEGs were significantly enriched in GO terms and metabolic pathways by Cytoscape v 3.9.1 software [24]. The GO terms or KEGG pathways with a Benjamini–Hochberg FDR *p*-value of 0.05 were considered significant when identifying the functional categories.

The protein–protein interaction (PPI) network between the DEGs was analyzed using the STRING database (http://string-db.org/, accessed on 21 March 2023), which includes direct and indirect protein associations [25]. Based on the results of the STRING analysis and the expression information of DEGs, the network diagram of DEGs was generated using Cytoscape software [24]. The topology scores of the nodes in the PPI network were calculated using the CytoNCA plugin v2.1.6 in Cytoscape v3.7.2.

### 2.5. Weighted Gene Co-Expression Network Analysis

Weighted gene co-expression networks were constructed using the WGCNA package v1.72-5 of R v4.3.1 [26]. The scale-free topology fit index of 0.8 was used to find an appropriate soft threshold power. The soft threshold was set to 12. The parameters setting for module classification were as follows: verbose = 5; mergeCutHeight = 0.02; and minModularSize = 50. Pearson correlation coefficients for relationships between the module and sample trait were calculated according to the ME values for modules. Modules with Pearson correlation coefficient > 0.5 and *p* < 0.05 were considered to be associated with ovarian weight and number of follicles. The modules significantly associated with traits were selected as key modules for PPI network analysis.

### 2.6. Quantitative RT-PCR Validation of mRNA Expression

Ten DEGs were randomly selected for quantitative real-time PCR (qRT-PCR) validation of the RNA-seq results. Total RNA was extracted from ovaries and SWFs by the Trizol method. The isolated RNA was reverse-transcribed into cDNA with the ALL-In-One 5X RT MasterMix kit (ABM, Vancouver, BC, Canada). qRT-PCR was performed on a CFX96 Real-time System (Bio-Rad, Hercules, CA, USA) as follows: a cycle at 95 °C for 30 s, followed by 40 cycles of 95 °C denaturation at 5 s and annealing at 60 °C for 30 s. The 10 μL qRT-PCR system included 5 μL of 2 × S6 Universal SYBR qPCR Mix (EnzyArtisan, Shanghai, China), 0.2 μL of each primer (10 μM), 1 μL of template cDNA, and 3.6 μL of ddH_2_O. The gene primer sequence was designed on NCBI and synthesized by Tsingke Biotechnology Company (Hangzhou, Zhejiang, China). The reference gene *GAPDH* was used to standardize the expression level of candidate genes. The expression levels of candidate genes were determined using the 2^−ΔΔCT^ method.

## 3. Results

### 3.1. Summary of RNA-seq Data

Transcriptome sequencing was performed on the ovaries and SWFs of six female geese at two different photoperiods. A total of 580,219,574 raw reads were obtained, of which 282,914,898 were from ovaries and 297,304,676 were from follicles. After excluding low-quality data, there were 280,152,902 clean reads from ovaries and 294,646,488 from follicles. Q20 and Q30 were greater than 97.87% and 94.01%, respectively. The GC contents of all samples ranged from 47.93% to 48.86%. Mapping to the reference genome of *Anser cygnoides*, the mapped read rate ranged from 82.55% to 86.42%.

### 3.2. Identification of Differentially Expressed Genes

A total of 25,860 expressed genes were detected, including 19,204 known gene names and 6656 unidentified named genes from de novo assembly with StringTie. In the ovary, 152 up-regulated and 35 down-regulated DEGs were revealed by comparison of SP with LP samples (Figure 1A, Appendix A). Among them, the most important potential genes for long photoperiods to affect the development of the ovary were uncovered, such as *SPP1*, *C6*, *MZB1*, *GP1BA*, *FCGBP*, *ZP1*, *SERPING1*, *TOP2A*, *JCHAIN,* and *PVALB*. Short photoperiods increase some ovarian gene expression compared to longer photoperiods, including *KLHL30*, *GATA5*, *SLC29A4*, *ISL1*, *PTCHD3*, *SNCG*, *RRAD*, *TEX36*, *FGF1*, and *RDH8*. In SWFs, 315 up-regulated and 133 down-regulated DEGs were discovered in the comparison of SP with LP samples (Figure 1B, Appendix A). Among them, the most interesting potential genes for long photoperiods affected the development of the SWFs, such as *SPP1*, *ANGPTL5*, *ALPL*, *ZP1*, *CHRNA4*, *SLC35E4*, *TOX*, *RASD1*, *IQSEC3*, and *ZNF469*. Compared to longer photoperiods, short photoperiods enhance the expression of several SWF genes, including *CLDN34*, *SLC16A8*, *OPRK1*, *SPATA16*, *NPPC*, *RFX4*, *DUOX2*, *SLC14A2*, *FGB*, and *COL10A1*.

### 3.3. Enrichment Analysis of Pathways

The functional enrichment of DEGs was analyzed separately for ovaries and SWFs under different photoperiods. The top 20 terms with high enrichment significance are displayed in Figure 2. DEGs of ovaries were significantly enriched in 117 GO categories and 12 KEGG pathways, such as signaling pathways regulating pluripotency of stem cells, respiratory system development, negative regulation of cytokine-mediated signaling pathway, phospholipid homeostasis, mesenchyme morphogenesis, regulation of smooth muscle cell proliferation, negative regulation of response to cytokine stimulus, smooth muscle cell proliferation, etc. (Figure 2A, Appendix A). A total of 448 DEGs in SWFs were significantly enriched in 688 GO categories and 46 KEGG pathways, which included extracellular matrix organization, extracellular structure organization, extracellular matrix, collagen-containing extracellular matrix, anatomical structure morphogenesis, extracellular matrix structural constituent conferring tensile strength, circulatory system development, intrinsic component of plasma membrane, extracellular matrix component, animal organ morphogenesis, embryonic morphogenesis, integral component of plasma membrane, endoderm formation, embryo development, protein digestion, and absorption, etc. (Figure 2B, Appendix A).

### 3.4. Protein–Protein Interaction Network

To reveal the relationship between these DEGs, protein–protein interaction analysis was performed based on the STRING database. In the ovaries, a total of 30 nodes and 71 edges were established. The prominent node genes of the PPI network were *EP300*, *RAC2*, *TLR4*, *NF1*, *JAK2,* and *TET2* (Figure 3A). These node genes are significantly enriched in 23 GO pathways and 3 KEGG pathways, such as negative regulation of leukocyte differentiation, cellular response to corticoid stimuli, regulation of NIK/NF kappaB signaling, and vascular-associated smooth muscle cell proliferation. A total of 20 nodes and 23 edges were identified in the PPI network of the SWFs, with the node genes being *SLC6A1*, *ETV4*, *DAZL*, *GDNF*, and *NPY* (Figure 3B). The node genes in the PPI network of the SWFs were significantly enriched in eight GO pathways, such as neuron uptake, animal organization formation, and RNA stabilization.

### 3.5. Weighted Gene Co-Expression Network Construction and Module Detection

There were significant differences in ovarian weight between the SP and LP groups, with the ovarian weight being 109.32 ± 50.86 in the SP group and 12.41 ± 12.89 in the LP group (Appendix A). The number of small white follicles between the SP and LP groups was significantly different. The number of small white follicles in the SP group was 47.00 ± 13.15, and that in the LP group was 71.88 ± 24.50 (Appendix A). To explore the modules involved in the regulation of ovarian weight and number of small white follicles, a total of 11,992 genes were obtained to build the weighted gene co-expression network. After determining the scale-free topological model and mean connectivity, a soft threshold of 16 was considered the best soft threshold, as the R^2^ of the scale-free network was greater than 0.8 (Figure 4A,B). After dynamic tree trimming, 31 co-expression gene modules were identified (Figure 4C).

To identify co-expression modules associated with reproductive performance in geese, we evaluated the relationship between ovarian weight, number of follicle traits, and the module eigengene (ME) (Figure 5). In total, only 2 modules were significantly positively correlated with ovarian weight, while 17 modules were significantly correlated with SWFs, of which 6 modules were significantly positively correlated and 11 modules were significantly negatively correlated. The purple module involving 873 genes was significantly correlated with both ovarian weight (R = 0.74, *p* = 0.009) and number of follicles (R = −0.78, *p* = 0.004). The correlation coefficients between blue, yellow, turquoise, pink, and light green modules and SWFs were 0.98, 0.96, 0.91, −0.88, and −0.84, respectively. Further screening of hub genes for purple, yellow, and steel blue modules was performed by PPI network analysis. In the purple module, a total of 40 nodes were established. The prominent node genes of the PPI network were *KDM6A*, *PIK3R2*, *HSP90AA1*, *FGFR3,* and *TNRC18* (Appendix A). A total of 52 nodes were identified in the PPI network of the yellow module, with the node genes being *SIL1*, *MRPS10*, *MRPS14*, *GFM1*, *MRRF,* and *MRPS9* (Appendix A). Due to the limited number of genes in the steel blue module, no node genes were identified by PPI network analysis.

Enrichment analysis was performed for genes in the purple, blue, yellow, turquoise, pink, and light green modules. Genes in the purple module were significantly enriched in 24 GO categories and two KEGG pathways, such as muscular septum morphogenesis, body morphogenesis, glioma, positive regulation of chemokine secretion, and regulation of chemokine secretion (Appendix A). There were 84 GO categories, including cellular protein localization, cellular macromolecule localization, organelle membrane, intracellular transport, and nucleotide binding, which were significantly enriched in the blue module (Appendix A). The yellow module was significantly enriched in 38 GO categories, such as mitochondrial matrix, mitochondrial part, mitochondrial gene expression, mitochondrial translation, and mitotic cell cycle process (Appendix A). In the turquoise module, genes were significantly enriched in 18 GO categories, including mitotic nuclear division, RNA binding, nuclear division, poly-N-acetyllactosamine biosynthetic process, and mitotic cell cycle process (Appendix A). There were 20 GO categories and one KEGG pathway significantly enriched in the pink module, including regulation of the transforming growth factor beta receptor signaling pathway, regulation of cellular response to transforming growth factor beta stimulus, antigen processing and presentation of peptide antigens, positive regulation of the apoptotic signaling pathway, and negative regulation of cyclin-dependent protein serine/threonine kinase activity (Appendix A). The light green module was only significantly enriched in three GO categories, including negative regulation of DNA replication, positive regulation of osteoblast differentiation, and cellular response to nerve growth factor stimulus (Appendix A).

### 3.6. Validation of the RNA-seq Data by Quantitative RT–PCR 

To verify the reliability of the RNA-seq data, 10 DEGs were randomly selected. *CPXM1*, *HSPG2*, *SGCD*, *MGLL*, and *OLAH* in SWFs and *IGF1*, *FCGBP*, *LDB3*, *NELL2*, and *RAB3IP* in ovaries were selected for validation by qRT-PCR. The trend in expression variation of the 10 genes in intergroup comparisons was consistent with the transcriptome sequencing results (Figure 6). Overall, the RT-qPCR analysis indicated the reliability of the RNA-seq results, confirming that the transcriptome sequencing data are reliable.

## 4. Discussion

The ovary is an important reproductive organ in adult poultry, including many different-sized follicles corresponding to the various follicular developmental stages [27]. RNA-seq analysis revealed 187 DEGs in ovaries, among which *BMPR2* was highly expressed in the SP group, while *BMP4* was highly expressed in the LP group. BMPs are members of the transforming growth factor β (TGF-β) superfamily, including BMP factors and ligands, receptors, and binding proteins. *BMPR-II* is a receptor for *BMP*, and it has been shown that *BMPR-II* expression is high in chicken follicular granulosa cells, suggesting that granulosa cells may be a major target for BMP action [28]. *BMP4*, *BMP7*, and *BMP15* bind to the type II receptor and subsequently trigger the smad protein signaling pathway to affect follicular development [29,30,31]. There are numerous reports that BMP family factors are expressed in the ovary of poultry and have important functions. In the present study, *BMPR2* was highly expressed in the ovary, probably by binding to *BMP* family ligands and contributing to ovarian and follicular development.

In both ovaries and SWFs of the SP group, we found that the *IGF1* gene was highly expressed. In birds as in mammals, various studies have shown that growth factors including *IGF1* (insulin-like growth factor-1) are involved in the regulation of follicular development [32,33,34,35]. The *IGF1* gene plays an important role in follicular growth and maturation. The *IGF1* can increase progesterone production and expression of STAR, CYP11A1, and 3βHSD in chicken granulosa cells at the preovulatory follicle stage [36,37]. Other reports have shown that *IGF1* enhanced in vitro progesterone production by granulosa cells [34,38,39,40]. In vitro studies have shown that the ability of granulosa cells to synthesize progesterone in preovulatory follicles of chickens increases with follicular maturation, reaching a peak at the mature follicle, and the same has been reported in geese [41,42]. In the present study, high expression of *IGF1* in the ovaries and SWFs may have promoted progesterone production in association with yolk deposition.

Follicular development is a complex physiological process, regulated by diverse genes and endocrine hormones [43]. Steroid hormones are important factors that regulate the normal growth of granulosa cells and oocytes and are involved in the process of follicular development, differentiation, and atresia [44]. The *SRD5A2* gene is significantly associated with steroid hormone biosynthesis [45]. Most studies on the *SRD5A2* gene are based on mammals, and there is a lack of research related to poultry. The *SRD5A2* gene was expressed in different tissues of goats with different lambing traits, and it may be related to the lambing traits of goats [46]. The steroid biosynthesis signaling pathway is a pathway associated with reproductive performance and may play a fundamental role in regulating ovarian function. In this study, the *SRD5A2* gene is highly expressed in SWFs, which may be useful for favoring follicular development. We found that the *G0S2* gene was highly expressed in SWF tissues of the LP group. The *G0S2* gene has been shown to be involved in the regulation of cell growth and development, and plays an important regulatory role in the cell cycle and cell proliferation. *G0S2* overexpression resulted in delayed oviposition and reduced egg production in female Japanese quails [47]. This may be due to the high expression of *G0S2* in adipose tissue, which inhibits its lipolytic activity in adipocytes upon binding to ATGL, thereby affecting avian yolk development [48,49,50,51,52]. Another study in poultry showed that *G0S2* regulated follicular granulosa cell development by modulating metabolic homeostasis in adipose tissue, which in turn affects egg production in laying hens [52]. 

We obtained an important purple module using WGCNA, and the purple module was significantly correlated with both ovarian weight and SWF number. Candidate genes *POSTN* and *VIPR2* were identified in the purple module. Periostin (POSTN) is an extracellular matrix protein involved in the remodeling of injured tissues. In recent years, it has been found that *POSTN* has a high expression in human ovaries [46]. The main function of *POSTN* is to participate in the adhesion and migration of ovarian epithelial cells [53]. A study confirmed that a new polymorphism in the *POSTN* gene was associated with egg production and egg weight or body weight in laying hens [54]. This indicates that *POSTN* can play a role in regulating ovarian development. In addition, overexpression of the *POSTN* gene in human pulmonary artery smooth muscle cells significantly increased the production of ET-1 and VEGF and promoted vascular smooth muscle production [55]. Another study indicated that *POSTN* can also regulate vascular endothelial cells and angiogenesis through the PI3K/AKT signaling pathway [56,57]. Morphological studies have shown that follicle development and corpus luteum formation are closely related to neovascularization. The denser and more developed the vascular network within the follicle and corpus luteum, the richer the supply of nutrients to the follicle and corpus luteum, and the better the follicle and corpus luteum develop. In contrast, sparse perifollicular and periluteal vascularization causes follicular atresia and luteal degeneration [58]. In the present study, the cell migration involved in sprouting angiogenesis and regulation of cell migration involved in sprouting angiogenesis pathways were significantly enriched GO pathways in the purple module. We hypothesized that *POSTN* could promote yolk deposition and follicle development by modulating angiogenesis. In a previous report, the *POSTN* gene was found to be associated with the proliferation and apoptosis of granulosa cells in chicken ovaries [59]. *VIPR2* (vasoactive intestinal peptide receptor 2) belongs to the VIP/PACAP type II receptors, also known as pituitary adenylate cyclase-activated polypeptide (PACAP) receptors. PACAP is a bioactive peptide transiently expressed in preovulatory follicles that stimulates ovarian function [60]. Transcriptome sequencing of small white follicles from low-laying Jilin black chickens and high-laying Lohmann Brown showed higher expression of *VIPR2* transcripts in Jilin black chickens, which presumably may block follicle growth and maturation in ovarian follicle development [61]. 

## 5. Conclusions

In conclusion, we constructed RNA sequencing libraries of ovaries and small white follicles under different photoperiods in Zhedong white geese. A total of 187 and 448 DEGs were identified in ovaries and SWFs with different photoperiods, respectively. GO and KEGG analyses suggested that response to signaling pathways regulating pluripotency of stem cells, phospholipid homeostasis, mesenchyme morphogenesis, regulation of smooth muscle cell proliferation, extracellular matrix organization, extracellular structure organization, and the extracellular matrix pathway may also play a key role in photoperiodic regulation of ovarian and follicle development in geese. With the identification of related genes by the WGCNA method and the functional analysis of differential genes by GO and KEGG, this study further provides a theoretical basis for the practical effects of enhanced light on egg-laying performance in Zhedong white geese.

## Figures and Tables

**Figure 1 animals-14-02747-f001:**
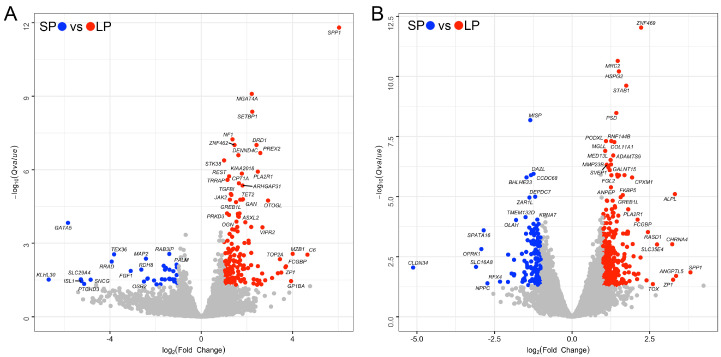
Volcano plot of DEGs between short-photoperiod group and long-photoperiod group. (**A**) Ovary. (**B**) Small white follicles.

**Figure 2 animals-14-02747-f002:**
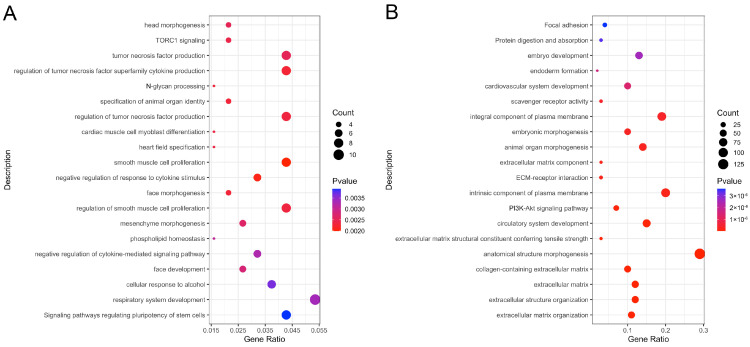
Pathway enrichment analysis of DEGs. (**A**) Ovary. (**B**) Small white follicles. The top 20 pathways with the most significant enrichment are displayed in the figure.

**Figure 3 animals-14-02747-f003:**
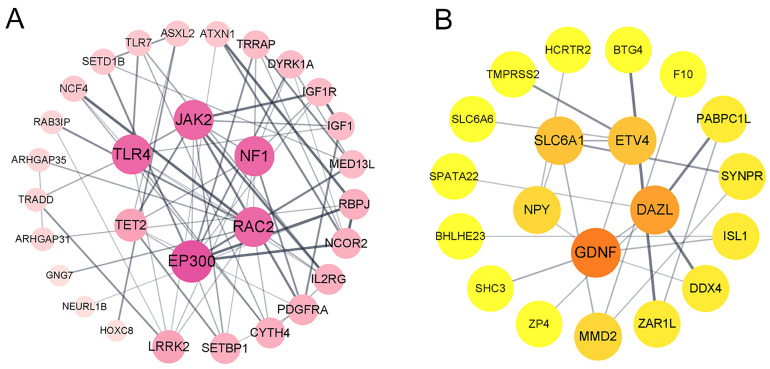
Significantly enriched PPI for DEGs of different photoperiods. (**A**) Ovary. (**B**) Small white follicles. Each node represents a gene, and the thickness of the line connecting two nodes indicates the strength of the protein interaction.

**Figure 4 animals-14-02747-f004:**
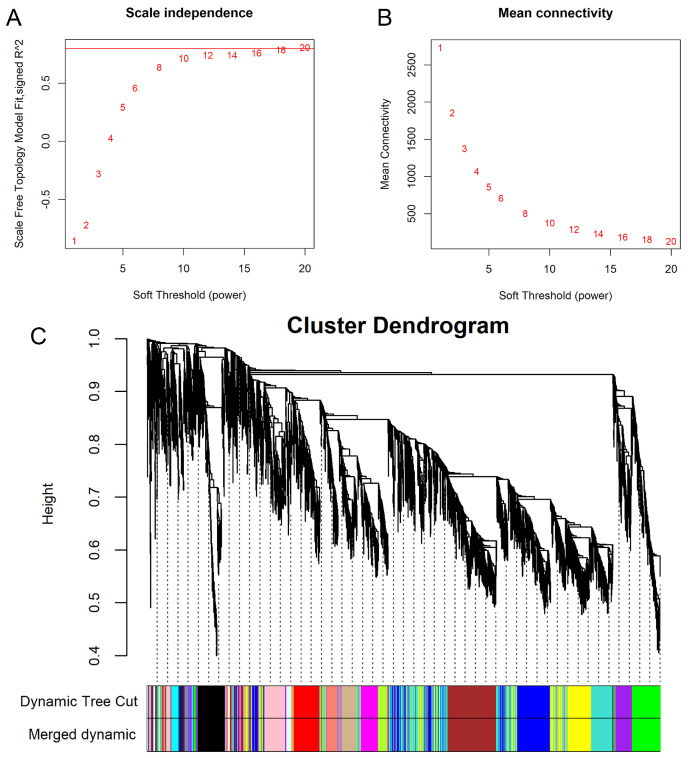
Weighted gene co-expression network analysis of the gene expression dataset. (**A**) Scale independence. (**B**) Mean connectivity. (**C**) Gene clustering tree. The upper part of the figure is the gene hierarchy clustering tree, and the lower part is the gene module. Different modules are represented by colors in the horizontal bar.

**Figure 5 animals-14-02747-f005:**
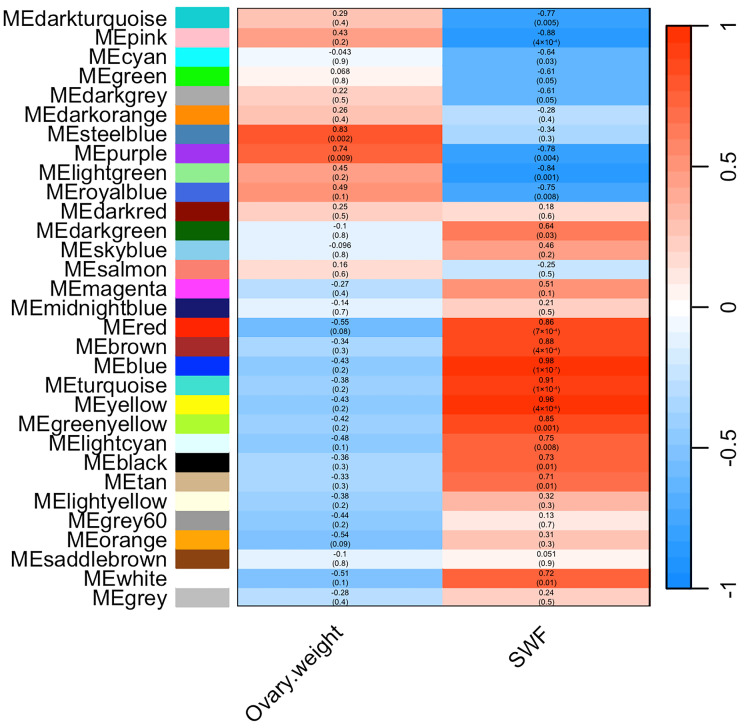
Heatmap of the correlation between modules and traits. Each column represents a trait, and each row denotes an eigengene for a certain module. The matching correlation and *p* value are included in each cell. The darker the color, the higher the correlation. Red represents a positive correlation; blue represents a negative correlation.

**Figure 6 animals-14-02747-f006:**
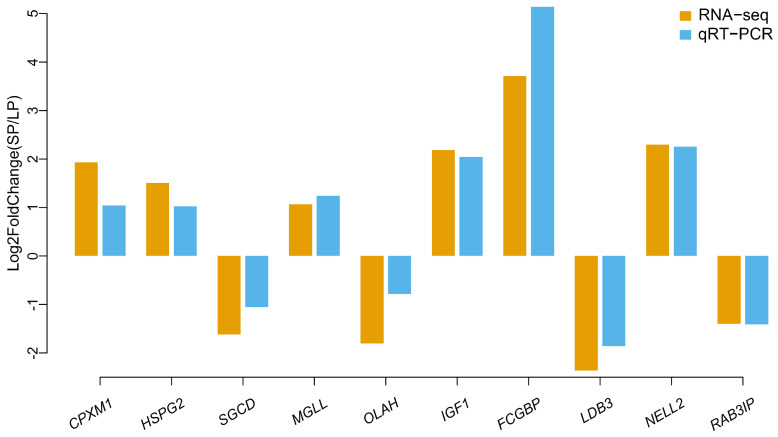
RT-qPCR validation of DEGs obtained by RNA-seq. Total RNA was extracted from the SWFs and ovaries and measured by qRT-PCR analysis. Relative expression levels were calculated according to the 2^−△△Ct^ method using *GAPDH* as an internal reference gene.

## Data Availability

The original data presented in this study can be found in online repositories. All sequencing data were submitted to the SRA database in NCBI with the accession number PRJNA1160041.

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
