# Peer review of "Effects of Different Photoperiods on the Transcriptome of the Ovary and Small White Follicles in Zhedong White Geese"

_animals, 2024, doi:10.3390/ani14182747_

Round 1
Reviewer 1 Report (Previous Reviewer 1)
Comments and Suggestions for Authors
My concerns have been well responsed. The manuscript can be accepted at present.
Reviewer 2 Report (Previous Reviewer 3)
Comments and Suggestions for Authors
Thank you for your responses to critiques. You have addressed them well.
This manuscript is a resubmission of an earlier submission. The following is a list of the peer review reports and author responses from that submission.
Round 1
Reviewer 1 Report
Comments and Suggestions for Authors
RNA-seq analysis identified many differentially expressed genes in ovaries and small white fol licles of different photoperiod groups in Zhedong white geese. With the identification of related genes by WGCNA analysis method and the functional analysis of differential genes by GO and KEGG, this study provided a better understanding of the molecular regulation in the photoperiod affecting goose reproduction. This research was well designed and can be accepted after minor revision.
1.The authors stated that the ovaries were weighed and the follicles of different grades were categorized and counted. I wonder whether the photoperiods affected the phenotype of these traits.
2. Abstract: Line 32: I suggest to change “increase” to “increases”. Discussion: Line 324: “showed that G0S2 regulates follicular granulosa cell development”-I suggest to change “regulates” to “regulated”.
3.Materials and Methods: Line 154: “The 10 μL qRT-PCR system included 5 μL of 2×S6 Universal SYBR qPCR Mix (EnzyArtisan, Shanghai, China), 0.2 μL of each primer (10 μM), 1 μL of template cDNA, and 4 μL of ddH2O.” I calculated and found that the reaction system was not 10 μL.
4. Line 174 and 180. Supplymentary files should be supplied to show more information for the DEGs.
5. Line 193 and 198. Supplymentary files should be supplied to show more information for the significant KEGG pathways.
6.Line 226: “A total of 11,992 genes were obtained to build the weighted gene co-expression network.” The amount of genes is so large, I wonder if there are too many genes in each module that make it difficult to search for key genes.
Reviewer 2 Report
Comments and Suggestions for Authors
The authors have conducted very relevant research that has fundamental and applied significance for the poultry industry.
Notes:
Figure 1 lacks a legend. Specify the color of the photoperiods.
Lines 83-84 "... and to explore the regulatory role of photoperiods on follicular development" should be corrected, since the article does not contain data on follicle development in birds in the experimental groups.
Reviewer 3 Report
Comments and Suggestions for Authors
The present manuscript reports on the effect of photoperiod on the transcriptome in small white follicles and ovarian tissue in geese. The investigators studied Zhedong geese, which are sensitive to photoperiod, and used artificial light to extend the photoperiod. The manuscript describes the qualitative study. The authors are unclear on the number of animals used: six at each photoperiod for twelve or three at each for six geese. Presumably, tissues were pooled across “the six female geese of two photoperiods” l 163, which removes animal variability. The authors have not provided a clear description of the design, two factors of tissue type and photoperiods, or the statistical analysis of the data. Although the manuscript is well-written for the most part, numerous misstatements detract from the readability. Singularly, this is not a particularly necessary publication. The results are valid but should be connected to another to contribute significantly to the literature.
Comments on the Quality of English LanguageL 20 rephrase - “respiratory system development” - misleading to think that ovaries breathe.
L 35 rephrase - “as well as embryo-related function in small white follicles” -follicles do not contain embryos.
L 93 incomplete sentence - Geese were raised
L 142 incomplete sentence
L 290 delete “are to”
L 299 STAR
L 300 stage
L 359 change lesser to small